# Targeting Bacterial Cell Division: A Binding Site-Centered Approach to the Most Promising Inhibitors of the Essential Protein FtsZ

**DOI:** 10.3390/antibiotics9020069

**Published:** 2020-02-07

**Authors:** Andrea Casiraghi, Lorenzo Suigo, Ermanno Valoti, Valentina Straniero

**Affiliations:** Dipartimento di Scienze Farmaceutiche, Università degli Studi di Milano, via Luigi Mangiagalli, 25, 20133 Milano, Italy; andrea.casiraghi@unimi.it (A.C.); lorenzo.suigo@unimi.it (L.S.); ermanno.valoti@unimi.it (E.V.)

**Keywords:** FtsZ inhibitors, bacterial cell division process, GTP-binding site, interdomain site

## Abstract

Binary fission is the most common mode of bacterial cell division and is mediated by a multiprotein complex denominated the divisome. The constriction of the Z-ring splits the mother bacterial cell into two daughter cells of the same size. The Z-ring is formed by the polymerization of FtsZ, a bacterial protein homologue of eukaryotic tubulin, and it represents the first step of bacterial cytokinesis. The high grade of conservation of FtsZ in most prokaryotic organisms and its relevance in orchestrating the whole division system make this protein a fascinating target in antibiotic research. Indeed, FtsZ inhibition results in the complete blockage of the division system and, consequently, in a bacteriostatic or a bactericidal effect. Since many papers and reviews already discussed the physiology of FtsZ and its auxiliary proteins, as well as the molecular mechanisms in which they are involved, here, we focus on the discussion of the most compelling FtsZ inhibitors, classified by their main protein binding sites and following a medicinal chemistry approach.

## 1. FtsZ

### 1.1. FtsZ and the Cell Division Process

Cell division is primarily coordinated by the multiprotein complex called the divisome. Several authors consider the whole bacterial cycle divisible into multiple steps, whose sequence leads to the formation of the divisome [1,2].

The trigger point of the process is the binding of FtsZ monomers to GTP, resulting in FtsZ polymerization into single-stranded filaments, following a defined ring-like structure: the Z-ring. This stage is regulated and controlled by multiple mechanisms, both positive and negative, since the correct localization of FtsZ polymerization is essential for obtaining appropriate daughter cells. It should take place at mid-cell. The Z-ring seems not to be composed of a single and circular FtsZ protofilament, but of cross-linked protofilament patches [2,3]. Indeed, the presence of several FtsZ protofilaments crosslinked was demonstrated only in vitro so far, i.e., when treated with divalent cations. Nevertheless, a definitive proof of in vivo protofilament lateral interactions is still missing.

After its formation, the Z-ring it is tethered to the cell membrane by a wide number of FtsZ-interacting proteins; for instance, in *Escherichia coli*, ZipA and FtsA tether the Z-ring through a specific conserved region of the protein, the *C*-terminal tail (or CTT), discussed later in this review.

Lastly, the maturation of the divisome is reached with the recruitment of several other proteins specifically involved in the synthesis of the cell wall (i.e., FtsI in *E. coli*). 

The final division is obtained through the constriction of the membrane, together with its invagination and the biosynthesis of peptidoglycan [2,3]. In particular, in *E. coli*, the essential septal transpeptidase FtsI moves directionally along the septum, promoting the biosynthesis of the peptidoglycan in concert with cell division [4]. Thus, FtsZ results as the main actor of this essential process, due to its peculiar functions: the ability to bind GTP, to polymerize into protofilaments, and to crosslink, coordinating the formation of the Z-ring, the crucial and limiting step of the cell division cycle.

### 1.2. FtsZ and Eukaryotic Tubulins

The homology between FtsZ and eukaryotic tubulins was exhaustively studied. The similarity among these proteins was proven at different levels; they share the ability to bind and hydrolyze GTP, as well as polymerize into protofilaments in a GTP-dependent manner [5]. Moreover, the GGGTGTG sequence of FtsZ is known to be also the signature motif of α-, β-, and γ-tubulins and is directly related to their ability to bind the GTP [6]. Even if FtsZ and tubulins demonstrated a high functional homology, their amino-acidic sequences largely diverge [7], making FtsZ a potential target for antimicrobial molecules with selective activity on prokaryotic cells.

### 1.3. FtsZ, Its Structure, and Main Inhibitor Binding Sites

Even if FtsZ functionality in the cell division process is widely conserved among most prokaryotic organisms, the structural conservation is strictly dependent on the specific protein portion. FtsZ could be divided into five main domains as reported in Figure 1: (1) the *N*-terminal subunit, which is unstructured and characterized by a poor conservation, has no particularly known functions; (2) the globular core, already largely studied, is highly conserved and it presents the important GTP-binding site; (3) the *C*-terminal linker, previously known as “variable spacer”, is characterized by a low degree of conservation and by a poor structuration; (4) the *C*-terminal Tail (CTT) is short and crucial for the interactions between FtsZ and several auxiliary proteins and for establishing lateral interaction between protofilaments in the Z-ring; (5) lastly, the *C*-terminal variable region (CTV) is able to promote lateral interactions in the absence of modulatory proteins [2,8]. 

More broadly, FtsZ can also be divided into the two main regions: The *N*-terminal domain, which contains the globular core and the GTP-binding site, and the *C*-terminal domain, including the CTT and the CTV. The two parts are connected by a central helix (H7) [9].

This second partition is especially convenient when considering the two main known binding sites for FtsZ inhibitors, the GTP-binding site and the interdomain site. These two protein sequences are hereunder considered and intensively examined, as we chose them as the criterion for the classification of the most interesting and promising FtsZ inhibitors developed in literature so far.

#### 1.3.1. GTP-Binding Site

As all the members of the tubulin superfamily, FtsZ polymerizes “head to tail” in a GTP-dependent manner. GTP–FtsZ binds at the bottom end of the filament completing the GTPase catalytic site. Hydrolysis of GTP decreases the interaction of the protein with the nucleotide and consequently results in the disassembling of the protofilament [1]. In *Methanococcus jannaschii*, a hyperthermophilic methanogen bacterium, six different FtsZ regions interact with GTP: loops T1–T4, the region for sugar binding, and the region for base recognition [10]. Considering the cruciality of the GTP binding ability of the protein, the GTP-binding site is widely conserved, as the whole globular core, not only among the different bacterial species, but also in eukaryotic tubulins. As a result, it became a target for designing broad-spectrum antibacterial agents.

Moreover, no drug-resistant mutants are reported in the literature so far, presumably because the GTP-binding site is essential for the correct recognition of GTP molecules. Amino-acid substitutions at this binding pocket might cause improper recognition of GTP molecules and, thus, hinder the normal GTP hydrolysis process, losing the energy source to drive the polymerization of FtsZ monomers.

#### 1.3.2. Interdomain Site

A wide number of FtsZ inhibitors interact with the interdomain binding site, a second interesting domain located in a long cleft between the *C*-terminal domain and the H7 helix of FtsZ. It spans over an extended surface and includes an area in a similar position to the taxol-binding site of tubulin. This interdomain cleft, located between the *N*-terminal and the *C*-terminal domains, opens and closes during the functional cycle of FtsZ as the protein switches between conformations with different propensity to polymerization in protofilaments. The site shows only a relatively limited conservation across FtsZs of different bacterial species. As a result, variations in the amino-acidic composition and site accessibility could account, at least in part, for the observed differences in the susceptibility of FtsZ from different bacterial strains to the same inhibitor or to inhibitors belonging to similar chemotypes [11]. While this variability may pose a challenge for the development of broad-spectrum antimicrobials, the specificity of this area and its structural divergence from the corresponding portions of tubulin may rule out potential toxicity concerns of inhibitors targeting the GTP-binding site, which shows a high homology between the two proteins.

In addition to FtsZ inhibitors targeting these two binding sites, some exceptions are reported. Among them, cinnamaldehyde, which is characterized by a minimum inhibitory concentration (MIC) of 7.5 mM and 3.8 mM vs. *E. coli* and *Bacillus Subtilis*, respectively, neither interacts with the GTP site nor with the interdomain site. In silico studies predicted cinnamaldehyde to interact with G295 and V208, involving also the T7 loop. [12]

### 1.4. FtsZ Inhibitors: How to Properly Evaluate Them

Currently, several FtsZ inhibitors are known in literature, both natural and synthetic. Several experimental procedures should be performed to properly ensure that FtsZ is the primary target of a putative inhibitor. All the possible assays, together with their aims and the most significant examples, were outstandingly summarized in the review of Kusuma and co-workers [13]. Following their guideline, the FtsZ inhibitory evaluation can be achieved with both in vitro and in vivo assays.

#### 1.4.1. In Vitro Assays

Saturation transfer difference NMR (STD-NMR), X-ray crystallography, surface plasmon resonance (SPR), and isothermal titration calorimetry (ITC) are the four most important methods to evaluate the direct interaction with FtsZ, even if further ones could be set up and performed. Among these assays, STD-NMR is commonly used to evaluate which are the moieties involved in the interaction with the target, while ITC assesses the binding constant of any compound for FtsZ. After having investigated the affinity, it is also important to evaluate if and how FtsZ enzymatic activity changes when treated with the xenobiotic. For this aim, the most commonly used assays are 90° angle light scattering, GTP-dependent polymerization, and polymerization sedimentation assays. Potentially, colloidal aggregation of the tested compounds could disrupt the three-dimensionality of FtsZ, resulting in antimicrobial effects independently from the specific inhibition of FtsZ activity. 

#### 1.4.2. In Vivo Assays

*In vitro* assays are very useful to evaluate direct interactions. Nevertheless, in this kind of experiment, several factors are not considered, such as cell membrane permeability, efflux pump activity, and several others. Usually, in vivo assays are recommended for a more accurate assessment of antimicrobial efficacy. Among them, the most common method is the examination of any morphological change; indeed, inhibition of the bacterial division system results in an abnormal lateral growth, leading to elongated, filamentous, or enlarged cells. This evaluation can be easily performed using phase-contrast microscopy and should be followed or preceded by other specific assays.

As a result, in order to confirm the binding of compounds with FtsZ and its consequent inhibition, any researcher should proceed using a multi-assay approach, which combines both in vitro and in vivo experiments (Figure 2). 

As suggested by Kusuma and co-workers [13], the first step to evaluate the cell division inhibition should be the phenotype evaluation. This quick and simple experiment allows easily excluding non-active compounds and, in contrast, could give an initial strong proof of division inhibition. 

Secondly, ITC, SPR, or other binding assays can directly evaluate in vitro FtsZ interaction, supporting the interaction with the target and giving additional information (i.e., K_d_). Lastly, enzymatic assays can measure if FtsZ activity is altered by the presence of the given compound. Positive results from all these approaches yield strong evidence of the inhibition of the cell division system consequent to the inhibition of FtsZ. The following steps should be the evaluation of the promising derivatives in an animal model, together with the study of their pharmacokinetics.

## 2. FtsZ Inhibitors

Hereafter, we present the most compelling FtsZ inhibitors discovered in the last decade, coming from various research groups (see Table 1). We decided to divide them considering the two binding sites and highlighting the best chemical classes. For each class, we deeply evaluate the most significant compounds, together with their antimicrobial results and the most significant details.

### 2.1. GTP-Binding Site Inhibitors

#### 2.1.1. Pyrimidines

Chan and coworkers started from an initial virtual screening to identify a novel class of FtsZ inhibitors, potentially useful for further chemical modifications. They screened a library of more than 20,000 compounds (both natural and synthetic), downloaded from Analyticon Discovery, for binding at the GTP-binding site of the *M. jannaschii* FtsZ (Protein Data Bank (PDB) 1W5B) [54].

No physicochemical property filtering was employed prior to docking, considering that natural products do not generally respect Lipinsky rules for permeation and absorption. From these calculations **N1** in Figure 3, 4-(((2*R*, 4*S*, 5*R*)-5-(2-methyl-6-(thiophen-2-yl)pyrimidin-4-yl)-quinuclidin-2-yl)methylcarbamoyl)butanoic acid arose as a promising derivative. They purchased it and tested it *in vitro*; the resulting mild activities on both *Staphylococcus aureus* American Type Culture Collection (ATCC) 29213 (449 μM) and *E. coli* ATCC 25922 (897 μM) and the ability to inhibit FtsZ GTPase activity (half maximal inhibitory concentration (IC_50_) = 317.2 μM) made **N1** the starting hit for sequential in silico optimization and structure–activity relationship (SAR) study, achieving a wide series of first-generation quinuclidines.

Among them, **N2** and **N3** emerged (Figure 3). **N2** showed more than 17-fold improved antibacterial activities (25 μM on *S. aureus* ATCC 29213 and 49 μM on *E. coli* ATCC 25922) and a 10-fold more potent GTPase inhibition (IC_50_ = 37.5 μM), whereas **N3** had half of **N2** inhibitory activities (MIC of 50 μM on *S. aureus* ATCC 29213 and 76 μM on *E. coli* ATCC 25922, and IC_50_ = 73.2 μM) but it is still promising. 

Moreover, **N2** was proven to strongly interfere with FtsZ assembly, without affecting tubulin polymerization [54], while **N3** was tested in combination with several β-lactam antibiotics. This study showed how methicillin and imipenem activities against methicillin-resistant *S. aureus* (MRSA) were strongly enhanced by **N3**, and how the activity of quinuclidine itself was also improved [55]. Similar results are in line with that accomplished with benzamide derivatives PC190723 [25] and TXA707 [26,27], better explained later. 

The same researchers then aimed at further improving antimicrobial activity and at simplifying the structural complexity of the quinuclidine scaffold, which, through its rigidity, could limit the interaction with the target. They moved to simpler amines, in lieu of the quinuclidine, such as cyclic amines, piperazines, and linear amines. The best results were achieved with a seven-membered homopiperazine ring, with a substituted benzyl group, and with the benzyl group having a bulky group at the *para*-position [56]. A few discrete candidates stood out and the most encouraging was **N4** (Figure 3), with enhanced antimicrobial activities toward *S. aureus* ATCC 29213 (8 μM) and nine clinically isolated antibiotic-resistant *S. aureus* strains (MIC ranging from 1 to 6 μM). However, neither **N4** nor these other second-generation pyrimidines exhibited inhibitory activity toward the growth of *E. coli* ATCC 25922. The researchers attributed it to a potential poor penetration into these bacteria, even though no further investigations were performed on mutated *E. coli*, as done for PC190723 and its prodrug TXA436 [28] and for the benzodioxane derivatives [29], described in a later paragraph of this work.

Nevertheless, Chan and coworkers studied the spontaneous resistance of **N4** in MRSA and evaluated that the rate of spontaneous resistant mutants was as low as <1 × 10^9^. They also highlighted the potential of **N4** for further animal studies, after having performed a preliminary measure of in vivo toxicity and efficacy, testing **N4** in a *Galleria mellonella* model [57]. Investigation on **N4** mode of action by STD-NMR and by biochemical assays confirmed the inhibition of *S. aureus* FtsZ. Specifically, NMR data indicated that the pyrimidine proton is in close relationship with the protein, while polymerization and GTPase assays showed how **N4** suppresses FtsZ self-polymerization, via disrupting FtsZ GTPase hydrolysis. Furthermore, exposure of *B. subtilis* cells to **N4** resulted in cell elongation and lack of the mid-cell foci. 

Recently, Fang and coworkers designed, synthetized, and evaluated novel 2,4-disubtituted-6-thiophenyl-pyrimidines (**N5**–**N8** in Figure 3), which were tested over a number of Gram-positive and Gram-negative bacterial strains (see Table 2), both sensitive and resistant to several antibiotics [14,15]. Their strategy in the design of these molecules was to keep the piperidine ring, which resulted as essential from the previously mentioned STD-NMR studies, and to substitute it with methyl and 4-pyridyl groups at position 2-, to investigate the steric effect at this position. The strong difference in antibacterial potency between **N5** and **N6**–**N8** revealed that the pyridyl group is crucial for the maintenance of a potent inhibitory activity.

**N5**–**N8** exhibited a stronger antibacterial activity against Gram-positive bacteria than Gram-negative strains; the author explained this behavior as a poor ability of these compounds to bypass the Gram-negative outer membrane.

Regarding Gram-positive bacteria, they showed promising MIC values on methicillin-resistant *S. aureus* (as MRSA: ATCC BAA-41, ATCC BAA-1717, ATCC BAA-1720, ATCC BAA-1747, ATCC 33591, ATCC 33592, ATCC 43300), as well as on vancomycin-resistant *E. faecalis* (ATCC 51575) and *E. faecium* (ATCC 700221). The activity of **N5** was lower than **N6**–**N8** across the whole panel of strains, suggesting a better outcome of the 4-pyridine ring in the interaction with the GTP-binding site, if compared to the methyl pendant.

However, all these third-generation pyrimidines proved to be bactericidal (minimum bactericidal concentration (MBC)/MIC ratio values of 1 or 2), to disrupt FtsZ polymerization, to inhibit the FtsZ GTPase activity, and to induce cell elongation. Furthermore, in addition to **N4**, **N5**–**N8** seemed not to induce the development of *S. aureus* resistant mutants and did not show any toxicity on human erythrocytes, suggesting a promising further development.

#### 2.1.2. Zantrins

In 2004, Margalit and coworkers performed a high-throughput protein-based chemical screening aimed at identifying potential molecules able to target FtsZ, specifically inhibiting its GTPase activity [16]. They started from 18,320 compounds and used a real-time, enzyme-coupled, fluorescent assay in a 384-well plate format.

Only 23 compounds were proven to directly inhibit FtsZ GTPase, and five of them (Z1–Z5, here called **N9**–**N13**, in Figure 4) were able to ~50% inhibit *E. coli* FtsZ GTPase activity at concentrations <50 μM. These compounds were named Zantrins because of their ability to act as FtsZ guanosine triphosphatase inhibitors. Specifically, **N9**, **N10**, and **N12** act as FtsZ assembly destabilizers, resulting in a diminution of protofilaments length and abundance, whereas **N11** and **N13** act as FtsZ assembly stabilizers, inducing protofilaments pairing and bundling. Moreover, **N11** acts as a strong stabilizer, while **N13** acts as a weaker one. 

The IC_50_s of **N9**–**N13** were evaluated against both *E. coli* and *Mycobacterium tuberculosis* FtsZ GTPases. The IC_50_s vs. *E. coli*, ranging between 4 and 25 μM, were significantly higher than the latter ones, whose values were one order of magnitude lower than the former. 

The Zantrins were tested over a wide range of Gram-positive and Gram-negative strains (see Table 3); only **N9** and **N10** showed an inhibitory activity on *E. coli*; the author hypothesized **N11**–**N13** as substrates of the resistance–nodulation–division (RND) efflux pump AcrAB, the major *E. coli* efflux pump. Therefore, **N9**–**N13** were evaluated over an *E. coli* AcrAB-null strain (DRC 39), showing an increased activity for **N10** and **N11**, whereas **N9**, **N12**, and **N13** maintained the initial inhibitory potencies.

**N9**–**N11** MICs showed promising activity versus *Shigella dysenteriae* and *Vibrio cholerae*. However, almost all the Zantrins were significantly more potent over Gram-positive strains, even on methicillin-resistant *S. aureus*. These initial data highlighted Z3 (**N11**) as having a drug-like structure and, thus, being a promising lead for further synthetic modifications [17,18].

A few years later, Anderson and researchers conducted preliminary SAR studies in **N11**, understanding that modifications at the halogen *para*-substitutions at the styryl portion negatively affected the activity [17]. Furthermore, the simplification of the diethylamino group into the dimethylamino one of **N14** resulted in a better IC_50_ (12 μM vs. 20 μM of **N11**). No MICs were found in the literature concerning **N14**, but its promising IC_50_ was the starting point for additional modifications by the same research group.

Nepomuceno reported a couple of years later an SAR study in which **N11** and **N14** were modified on the benzoquinazoline core, replacing the fused benzene with groups with different steric and electronic properties, on the 4-chlorostyryl fragment, substituting it with several isosters, or on the amino ethyl side chain, modifying one or both the nitrogen atoms or evaluating positively charged amines [18].

The best results were achieved with **N15** (IC_50_ = 24 μM), featuring a 6-methyl-quinazoline instead of the benzoquinazoline ring, and with **N16** (IC_50_ = 24 μM), a trifluoroacetate benzoquinazoline ammonium derivative. Any alteration in the 4-chlorostyril fragment resulted in the complete loss of inhibitory activity (IC_50_ > 128 μM).

Zantrins were always recognized to bind the GTPase binding site; nevertheless, a few months ago, Sogawa and coworkers deeply analyzed **N11** and **N14** in their binding properties with *M. tuberculosis* FtsZ [19].

They precisely evaluated the interactions of **N11** and **N14** with FtsZ at an electronic level, using ab initio fragment molecular orbital calculations, and they searched for binding sites all over the protein sequence. Their study suggested that the binding site is in the vicinity of the H6/H7 loop and that it is distinct from the GTP-binding site of *M. tubercolosis* FtsZ. These novel results suggested that further analyses should be performed, such as STD-NMR or co-crystallization, in order to permanently confirm them.

#### 2.1.3. Chrysophaentins

In 2010 Plaza and coworkers evaluated marine natural products as a potential source of FtsZ inhibitors, thanks to their peculiar chemical structures and considering their strong antimicrobial activities [20].

They prepared a methanol extract of the chrysophyte alga *Chrysophaeum taylori* and evaluated the inhibition activity on the growth of *S. aureus*, MRSA, *E. faecium*, and vancomycin-resistant *E. faecium* (VREF).

Eight macrocycles were isolated and characterized by NMR and MS, establishing their chemical structures. Further antimicrobial evaluation, enzymatic assays, transmission electron microscopy (TEM), STD-NMR, and molecular docking were performed. All these data highlighted chrysophaentin A (**N17**, see Figure 5) as a valuable FtsZ inhibitor, able to inhibit the growth of all the bacterial strains (see Table 4), thanks to its ability to block both FtsZ GTPase activity and polymerization. STD-NMR studies gained insight into its binding mode to FtsZ, identifying the GTP-binding site as its binding pocket.

The same research group developed an SAR study from **N17**, including natural and synthetic chrysophaentins, aiming at simplifying its structure and enlarging the *S. aureus*-tested strains, including clinical methicillin-resistant (MRSA) and multidrug-resistant (MDRSA) strains and clinical isolates (UAMS-1 and CA-MRSA USA 300-LAC). **N18**, a synthetic fragment of **N17**, showed comparable antimicrobial results, even if slightly lower than its parent molecule [21].

Using GTPase and competition assays, together with NMR and fluorescence data, they demonstrated that **N17** and **N18** inhibit both the GTPase and the polymerization activities of the protein, disrupting FtsZ assembly and, thus, the Z-ring in live bacteria [22].

#### 2.1.4. GTP Analogues and Derived Synthetic Inhibitors

Nucleotide derivatives were also evaluated as FtsZ inhibitors, able to interfere with FtsZ polymerization. In this context, a huge study was that performed by Läppchen and Andreu and their research groups. They started designing and synthetizing a structurally diverse series of 8-substituted GTP analogues [58,59,60] and investigated their effect on both FtsZ and tubulin, calculated their ability to interfere with FtsZ polymerization and GTPase activity, and evaluated the antimicrobial activity of their prodrugs.

BrGTP and MeOGTP, named **N19** and **N20** in Figure 6, were the most potent GTP analogues. The former was the first to be discovered [59]; it was proven to inhibit both *E. coli* FtsZ polymerization (IC_50_ = 37 μM) and its associated GTPase activity (IC_50_ = 60.2 μM). 

**N20**, designed a few years later [60], was even more potent, with an IC_50_ on FtsZ polymerization of 10 μM and a similar IC_50_ on GTPase activity (15 μM).

**N19** and **N20** stabilities to hydrolysis were calculated by HPLC and resulted to be less than 3%; furthermore, their binding affinities to FtsZ were determined using a method developed by the same research group [61]. The results suggested that the inhibitory potencies are strictly related to the binding affinities. Moreover, both **N19** and **N20** potently promoted tubulin polymerization and assembly, even more strongly than GTP itself. A few years later, the same researchers combined NMR experiments, biochemical assays, and molecular modeling to determine the conformations of **N19** and **N20**, both free and bound to FtsZ, using two different FtsZ proteins [62]. Both the GTP analogues generated important modifications at the interface between FtsZ monomers, in terms of size, shape, and electrostatic surface. 

Finally, Huecas and collaborators further elucidated the inhibition mode of **N19** and **N20** using a *mant* fluorophore-based assay [63]. Despite the interesting inhibition profile of **N19** and **N20**, none of their prodrugs resulted as having acceptable antibacterial activities [58]. This lack of antimicrobial potency was attributed to the probable poor penetration across the bacterial cell envelope and prompted the researchers to the evaluation of smaller molecules, targeting the same binding site without having a nucleotide structure.

They evaluated compounds from literature, conducted a virtual screening campaign, and tested in-house compounds, after a docking evaluation on the *B. subtilis* FtsZ GTP-binding site [23]. The most encouraging molecules were **N21**–**N23** (Figure 6), previously described as anticancer agents.

All three derivatives of this first generation of molecules showed high affinity for FtsZ; **N21** had a k_b_ of 4.3 × 10^5^ M^−1^ while **N22** and **N23** constants were more than three times higher (k_b_(**N22**) = 1.5 × 10^6^ and k_b_(**N23**) = 1.3 × 10^6^). They resulted in an inhibition of the growth of Gram-positive strains, with mild (**N21** and **N22**) to good (**N23**) values (see Table 5). Unfortunately, the activity on Gram-negative strains was scarce to null for all three compounds, and their inhibition mode on FtsZ function was not well elucidated.

Therefore, the researchers moved to a second generation of compounds [24]. They started from the chemical structures of **N21** and N**24**, a different in-house compound not published before including a 3,5-biphenyl core instead of the 1,3-naphtalene one of **N21**.

In this SAR study, the gallate core hydroxy groups were replaced by methoxide ones or the number of the substituents was reduced. They also eliminated one of the polyhydroxybenzoate moieties and removed or replaced the ester spacers. They designed and synthetized more than 30 derivatives, which were evaluated for their solubility in aqueous buffer, for their ability to bind the GTPase binding site, for their binding specificity, and for their antimicrobial activity on Gram-positive and Gram-negative strains.

**N25** and **N26** (Figure 6) were identified as compounds able to specifically inhibit FtsZ with high affinity and selectivity over the inhibition of tubulin polymerization, interfering with bacterial cytokinesis, disrupting cell viability through modification of FtsZ assembly and hampering cell division. Their antibacterial activity was promising toward Gram-positive bacteria, including multidrug-resistant strains. 

### 2.2. Interdomain Site Inhibitors

#### 2.2.1. Benzamides

Benzamide-based inhibitors form the most well studied and numerous group of FtsZ inhibitors reported so far. They include compounds with considerable structural diversity, organized around three main structural features (see Figure 7): A benzamide core, which is maintained through the whole series and is fundamental for activity. The 2,6-difluoro substitution was proven to increase activity. The benzamide group interacts via hydrogen bonds with specific residues in the T7 loop of FtsZ (e.g., Val207 and Leu209 in *S. aureus*) which are conserved across different species [11];An alkylenoxy or alkylenamine linker region of different lengths;A variable terminal region including either an alkyl chain (functionalized or non-functionalized) or a heterocyclic moiety, which is accommodated in a narrow, deep, and hydrophobic cavity inside the interdomain cleft. In general, affinity is promoted by the possibility to form additional interactions and is strongly limited by the hydrophobic nature and steric constraints of the binding site. A wide variety of groups of different size, conformation, and electronic properties were evaluated, and this variability is likely at the source of the different pharmacological and physicochemical profiles of the different inhibitors.

The original interest in this class was sparked by the inhibitory activity of 3-methoxybenzamide (**3-MBA, I1**) on the growing rate of *B. subtilis* [30]. Despite the weak on-target antimicrobial activity (MIC = 26490 μM), 3-MBA was an attractive starting point for fragment-based drug design, due to its low molecular weight and its ability to penetrate cell membranes. A medicinal chemistry program carried out by Prolysis, Inc. (now Biota Ltd.) explored the SAR of the starting molecule in order to ameliorate on-target activity and drug-like properties. In a first phase, the effect of substituents on the benzamide ring was evaluated, with the identification of the 2,6-difluoro substitution pattern and the homologation from methoxy to ethoxy as positive for activity. Further elongation of the alkyl chain led to a first lead compound, 2,6-difluorononyloxybenzamide (**DFNBA**, **I2**), with greatly increased antibacterial activity (MICs of 0.125 ug/mL on *B. subtilis* and 0.5 ug/mL on *S. aureus*) [31]. Interestingly, the antimicrobial potency was retained in methicillin-resistant *S. aureus* (MRSA) strains. The improvement of the suboptimal drug-like properties of DFNBA was the object of a second phase of the program, in which the long alkyl chain was systematically substituted with different heterocycles in order to decrease logP, the number of rotatable bonds, and the binding to plasma proteins, as well as to introduce potential sites for additional hydrogen bonds. The best results were obtained with thiazolopyridines and, specifically, the 6-chloro derivative **PC190723** (**I3**) provided the best combination of bactericidal activity, metabolic stability, and overall drug-like properties [32]. In general, hydrophobic substitutions on the heterocyclic system appear to be highly favored in terms of activity. In a different study [33], the same group proceeded with the target validation for PC190723, demonstrating the direct and concentration-dependent inhibition on the GTPase activity of FtsZ through the development of specific assays. Furthermore, promising in vivo activity on murine models of infection was shown, with an impressive 100% survival of mice inoculated with a potentially lethal dose of *S. aureus* compared to 0% survival in the non-treated controls. The biological evaluation of PC190723 on a broader set of bacterial strains revealed a consistent antimicrobial potency across several *Staphylococcus* species (including multidrug-resistant *S. aureus* strains) and substantial inactivity against a mixed panel of different Gram-positive and Gram-negative strains (see Table 6). According to existing data, PC190723 is generally considered to have bactericidal action. It binds selectively to the interdomain cleft of a specific conformation of FtsZ in which the C-terminal domain is rotated up and away from helix H7 and the *N*-terminal domain. This conformation decreased assembly cooperativity with higher propensity to aggregation, and PC190723 is considered to act as a FtsZ polymer stabilizer, with a mechanism of action similar to the stabilization of eukaryotic tubulin polymers induced by taxol [34]. Co-crystallization with *S. aureus* FtsZ located the binding site within a deep cleft formed by the *C*-terminal half of the H7 helix, the T7 loop, and the *C*-terminal four-stranded β-sheet [25,35]. PC190723 is the most well characterized FtsZ inhibitor to date, and the thorough evaluation of its pharmacological and physicochemical profile highlighted some potentially critical issues for its development and deployment as an innovative antimicrobial into clinic. Specifically, it generates resistant mutants with a significantly high frequency and, given the size constraints of the binding pocket, this issue may be difficult to address through chemical modifications alone. It is considered to be best suitable for combination therapies and an interesting synergistic effect with β-lactam drugs was reported. When co-administered with imipenem, PC190723 was able to restore β-lactam susceptibility in MRSA clinical isolates and murine infection models and, in turn, imipenem markedly lowered the frequency of PC190723-resistant mutants [25]. Only two (G196A and N263K) of the six spontaneous *S. aureus* mutant isolates are devoid of major morphological and/or growth alterations. Interestingly, G196 mutants (which account for 70–75% of all isolates) retain sensibility to smaller 3-MBA, suggesting that resistance could originate from the inaccessibility of larger benzamides to the binding pocket [36]. A second critical issue lies in the hydrophobicity of PC190723 (ClogP = 2.64) and in the consequent poor water solubility and potentially problematic formulation, especially in terms of oral availability. **TXY436** (**I4**), an *N*-Mannich base prodrug derivative of PC190723, is associated with a 100-fold increase in solubility compared to the parent molecule when formulated in an acidic (pH = 2.6) citrate aqueous solution compatible with in vivo administration. TXA436 is converted in PC190723 with suitable kinetics at physiologic pH and retains complete on-target potency against both MRSA and methicillin-sensitive *S. aureus* (MSSA). Moreover, TXA436 solutions are suitable for both intravenous and oral administration (bioavailability of 73% in mice), as it is efficacious in murine models of infection and shows minimal toxicity to mammalian Vero cells [37]. In a similar way, **TXY541** (**I5**), a methylpiperidine 4-carboxamide prodrug of PC190723, is 143 times more soluble than its parent molecule in orally available acidic citrate formulations and retains its bactericidal antistaphylococcal activity, with minimal toxicity on mammalian cells and borderline frequency of resistant mutants [38]. A third potentially critical feature of PC190723 and its prodrugs involves the observed marked differences in susceptibility across different bacterial species. PC190723 MICs (Table 6) indicate an optimal activity against Gram-positive *B. subtilis* and *B. cereus* and numerous strains of *Staphylococcii*, while it appears to be practically inactive on Gram-positive *Enterococcii* and *Streptococcii* and Gram-negative *E. coli*, *P. aeruginosa*, and *Haemophilus influenzae*. The original developers identified residue 307 in *S. aureus* FtsZ and corresponding residues in other species as determinants for species-specific susceptibility. This key position at the entrance to the narrow binding pocket is occupied by a valine in sensitive species, while non-sensitive species feature larger polar residues (arginine or histidine) in corresponding positions, which could partially block the access to the binding site and reduce FtsZ inhibition. To support this hypothesis, the authors reported increased MICs in *B. subtilis* mutant V307H strain for 3-MBA and PC190723 (two-fold and four-fold, respectively) [33]. In a later study [39], Kaul and coworkers further elaborated on the topic and postulated that Glu34 and Arg208, conserved in non-susceptible *Enterococcii* and *Streptococcii* (corresponding to His33 and Val207 in *S. aureus* and Gln33 and Val307 in *B. subtilis*), could form an intramolecular salt bridge, which could in turn interfere with the binding of PC190723 and TXY541 to the target binding site. The relevance of steric hindrance and binding site accessibility across species was further supported by the in silico comparison of FtsZ from different microorganisms. Kusuma et al. [11] reported that, in the context of a general conservation of FtsZ across species, staphylococcal FtsZs show closer homology among each other compared to FtsZs from other species. One key difference lies in the degree of curvature of the H7 helix, which in turn determines the size and shape of the binding cleft and, more specifically, the size of the cleft opening, which is significantly larger in *Staphylococcii* compared to *B. subtilis*, *M. tuberculosis*, *Aquifex Aeolicus*, and *P. aeruginosa*.

Biochemical assessment of the effects of PC190723 on the GTPase activity and the polymerization of isolated FtsZ showed filament morphology alterations and increased bundling only for *S. aureus* and *B. subtilis* FtsZ, while FtsZ from *E. coli* was identified as non-susceptible. However, the increase in GTPase activity and HPLC analyses supported the binding of PC190723 on *E. coli* FtsZ [40]. Such binding was later demonstrated through fluorescence anisotropy studies, and PC190723 (delivered as TXY436) was shown to interact with a binding site on *E. coli* FtsZ similar to the one on *S. aureus* FtsZ with an approximately nine-fold lower affinity [28]. In the same work, PC190723 was found to inhibit the polymerization of *E. coli* FtsZ in a concentration-dependent manner, suggesting a different mechanism of action from the one observed in *S. aureus*. Even more interestingly, the susceptibility of *E. coli* to the compound was restored after the genetic or pharmacological inactivation of RND efflux pump AcrAB, with MICs shifting from >155 µM (practically inactive) to 19.4 µM. Similar effects were also observed in Gram-negative *Acinetobacter baumannii* and *K. pneumoniae*, with MICs going from >155 µM to 38.8 µM and 19.4 µM, respectively. These findings support the idea that PC190723 is a substrate of AcrAB and that the intrinsic resistance of some wild-type Gram-negative bacteria may be at least in part due to the activity of this efflux system, as opposed to being the exclusive result of insufficient affinity and/or limited accessibility to the binding site. Given the widespread distribution of AcrAB and other RND efflux pumps across Gram-negative species, these results shine a new light on species-specific susceptibility to PC190723 (and related benzamides) and open an interesting avenue for association therapies for highly clinically relevant pathogens. Despite its improved bioavailability profile, TXY436 shows suboptimal in vivo pharmacokinetic properties. Cytochrome P450 (CYP450)-mediated dechlorination/oxygenation of TXY436 activation product PC190723 results in rapid elimination and a short half-life. The replacement of chlorine with metabolically stable, electron-withdrawing, and hydrophobic –CF_3_ in methylpiperidine 4-carboxamide prodrug **TXA709** (**I6**) successfully increased the half-life and reduced the high doses required for in vivo oral activity in mice. TXA707, the activation product of TXA709, is able to rescue sensitivity of MRSA to various β-lactams [26,27]. As previously observed with PC190723 [25], the synergistic effects result from inhibitor-induced delocalization of FtsZ, which in turn alters the localization of penicillin-binding proteins (PBPs) and, consequently, causes a reduction in the required concentration of the β-lactam. Originating from the same medicinal chemistry program, benzyl bromo-oxazole 2,6-difluorobenzamide **I7** is four- to eight-fold more potent than PC190723 and is the most potent antistaphylococcal benzamide so far, with MICs of 1.40–2.81 μM against several *Staphylococcus* strains [41]. The *R*-(+) form is the most potent and is associated with a 133-fold decrease in MIC compared to the *S*-(−) form. Interestingly, it retains some activity against G196A *S. aureus*, the most common resistant mutant induced by PC190723, with a 32-fold decrease in susceptibility (MIC = 7.9 μM). In addition, the frequency of resistant mutations is comparable to that observed with PC190723, but the most frequent isolates show compromised fitness *in vitro*. A compelling explanation for the improved resistance profile was provided in a later work on close analogue **TXA6101** (**I8**) [42], which has antistaphylococcal potency comparable to **I7** and is active on TXA707-resistant G193D and G196S mutants (MIC = 2 μM). In TXA707, the five- and six-membered rings (thiazole and pyridyl) are fused together and have no rotational flexibility. Conversely, in TXA6101 the five- and six-membered rings (5-bromo oxazole and phenyl) are not fused and are connected by a freely rotatable single bond. This enhanced flexibility reduces steric clashes with more hindered Asp193 and Ser196 and expands activity to TXA707-resistant mutants, supporting the relevance of conformation for binding site accessibility and, ultimately, antimicrobial potency and susceptibility. In the same work, the role of the bromine atom on the oxazole ring (not present in TXA707) was investigated and was considered as highly positive for affinity and activity, thanks to extensive additional interactions with hydrophobic amino-acid residues on the C-terminal subdomain of *S. aureus* FtsZ. 

In the years following the discovery of PC190723, the work of several groups significantly expanded the structure–activity relationships of benzamide-based FtsZ inhibitors, mainly through modifications in the heterocyclic portion of the molecule. In a series of papers, Valoti and coworkers focused on the exploration of the 1,4-benzodioxane scaffold and on the effects of substituents and stereochemistry on FtsZ inhibition. In particular, the role of the 1,4-dioxane oxygen atoms was evaluated and they (especially O-1) were identified as crucial for the maintenance of high antibacterial potency, most likely thanks to the formation of additional hydrogen bonds in the binding site. The replacement with different hydrogen bond acceptor groups (such as *N*-methyl) proved to be detrimental, possibly due to steric clash in the limited space of the binding cleft [43]. Moreover, the effect of substituents on the phenyl moiety of benzodioxane was studied, and substitutions in position 7 (*meta* to O-1) resulted as the best in terms of activity. From the screening of numerous hydrophilic/hydrophobic, differently hindered, and bioisosteric substitutions in position 7, **I9** (7-chloro) [44] and **I10** (7-carboxymethyl) [45] emerged as the most promising derivatives, with MICs on *S. aureus* of 0.7 μM and 1.6 μM, respectively. In accordance with previous results on the binding of benzamide FtsZ inhibitors, hydrophilic and bulky substituents are disfavored, leading in some cases to dramatic loss in activity. Interestingly, the evaluation of single enantiomers identified the (*S*) form of **I9** as the most active, being four times more potent than the (*R*) form. Both compounds show minimal cytotoxicity on mammalian cell with very high therapeutic indexes. The on-target activity of **I9** and **I10** was confirmed with FtsZ GTPase activity and polymerization activity assays, which supported a polymer-stabilizing mechanism of action for this class of inhibitors, similarly to that previously observed for PC190723 [43]. The morphological analysis of treated bacteria using TEM showed swelling and septum malformations consistent with inhibitory activity on FtsZ. In recent work [29], docking simulations corroborated the binding of this class of compounds to the interdomain site of *S. aureus*, maintaining the key interactions described for PC190723. In addition to the promising activity on MSSA and MRSA strains, **I9** showed activity against *M. tuberculosis* (MIC = 22.5 μM) and both vancomycin-susceptible (VSE) and -resistant (VRE) *E. faecalis*, with MICs of 72 μM. On the other hand, both **I9** and **I10** are practically inactive on various strains of *E. coli*, but the activity of **I10** was partially recovered (MIC = 42.2 μM) in *E. coli* N43, a mutant strain lacking the AcrA component of the AcrAB efflux pump. In line with that previously observed for TXA436, this result suggests that this class of compounds may be substrates for AcrAB and that wild-type *E. coli* non-sensitivity to **I9** and **I10** (and related compounds) may not be exclusively due to the complete non-susceptibility of *E. coli* FtsZ. The effect of the pharmacological inhibition of AcrAB on the activity of these compounds is yet to be evaluated and may represent an interesting option for combination therapies. 

The role of different heterocycles in the SAR of heteroaryloxy-benzamide FtsZ inhibitor was further investigated by Bi et al. with the development and antimicrobial evaluation of a series of isoxazole-benzamide analogues [46]. Starting from the activity data of a first series of isoxazol-3-yl derivatives, the authors relied on docking simulations to identify affinity determinants and guide later structure-based optimization. In addition to confirming the expected hydrogen bond and hydrophobic interactions, simulations highlighted a potential additional ion–dipole interaction between the nitrogen of isoxazole and a nearby negatively charged residue (Asp199). By switching the relative positions of the N and O atoms, N gets closer to Asp199 and becomes able to establish the additional interaction. As a result of the increased affinity, isoxazol-5-yl derivative **I11** is two- to eight-fold more potent than the corresponding isoxazol-3-yl compound and is the most potent analogue in the series, with MICs of 0.04 μM on *B. subtilis*, 0.08 μM on *B. pumilus*, and 5.2 μM on *S. aureus*. The FtsZ inhibition effect of **I11** was supported by the concentration-dependent stimulation of *B. subtilis* FtsZ polymerization and by microscopic observation of the filamentation of treated *B. pumilus*. TEM visualization confirmed the increase in *B. subtilis* FtsZ protofilament size and thickness and increased bundling of protofilaments. The 3-(4,5-dimethylthiazol-2-yl)-2,5-diphenyltetrazolium bromide (MTT)-based assays revealed minimal cytotoxicity on HeLa cells, and **I11** showed interesting preliminary in vivo efficacy results, being able to significantly reduce the bacterial count in a murine model of systemic MRSA infection. In a more recent work, the same research group designed and synthesized six series of 1,3,4-oxadiazol-2-one, 1,2,4-triazol-3-one, and pyrazolin-5-one derivatives and evaluated their antimicrobial activity [47]. The 1,3,4-oxadiazol-2-one-based **I12** was identified as the most potent compound with an interesting bactericidal action against *B. subtilis* (MIC = 0.3 μM), *B. pumilus* (MIC = 2.3 μM), and various strains of *S. aureus* (MIC = 1.2–2.4 μM), despite substantial inactivity against *E. coli* and *P. aeruginosa* (Table 7). As in the previous study, the on-target effect was confirmed with *B. subtilis* FtsZ light scattering polymerization assays and morphometric observation of treated *B. pumilus*. Docking simulations indicated binding in the same pocket as PC190723 and congeners, with the amide and the ethereal oxygen involved in hydrogen bonds and the heterocyclic moiety accommodated in the hydrophobic channel. Prompted by the activity of **I12**, the authors proceeded with the evaluation of a large number of structural modifications. Conversion of the phenyl into benzyl with the introduction of an extra methylene group led to decreased activity, highlighting once again the steric constraints of the narrow binding site. Accordingly, a third series of analogues with more flexible ethyloxy linkers showed a more limited decrease in activity (MICs 0.28 to 9 μM). Modifications at the oxazolone core in the attempt to reduce the rate of degradation by hydrolysis led invariably to detrimental effects on activity, probably due to steric hindrance or to increased hydrophilicity. Stability studies of **I12** revealed good liver microsome stability profile and rapid biodegradation in mouse plasma *in vitro*. The short half-life (2.5 h) may not be optimal for further drug development, and ways to improve the biostability of **I12** are still under research.

Interestingly, non-heterocyclic derivatives were the object of study as direct analogues of DFNBA (**I2**), in some cases with interesting results in terms of antimicrobial activity. After the screening of a number of modifications of the 3-alkoxy sidechain, Bi et al. identified branched alkyl derivative **I13** [48] as the most potent compound with MICs of 0.88 μM on *B. subtilis* and 3.5–31.0 μM on various strains of *S. aureus* (Table 7). On the other hand, **I13** was practically inactive on *S. pyogenes*, *S. pneumoniae*, *E. coli*, and *P. aeruginosa*. In accordance with the general model for the interaction, hydrophobic alkylhalide chains of suitable length were tolerated, while bulkier and rigid cycloalkyls led to decreased activity. Lui et al. reported the interesting activity of nonylaminobenzamide **I14** [49] with MIC = 5.8 μM on *S.aureus* and substantial inactivity on *E. coli*. Time–kill curves suggested a bacteriostatic action and, interestingly, biochemical and morphological evaluations indicated reduced GTPase activity and inhibition of *S. aureus* FtsZ polymerization and delocalization of the Z-ring, supporting a different mechanism of action from polymer-stabilizing PC190723 and analogues. **I14** also showed promising results as a possible adjuvant antimicrobial in association with different classes of β-lactams. Combinations with methicillin, cloxacillin, amoxicillin, cefuroxime, and meropenem were assayed against a panel of 28 clinical MRSA strains and showed synergistic effects, possibly with bactericidal effects. 

#### 2.2.2. Berberine Analogues and Related Quinolinium Compounds

Berberine (**I15** in Figure 8) is a plant alkaloid with moderate antimicrobial activity [64] and FtsZ inhibition properties [65]. Sun et al. performed docking studies to identify the berberine-binding site on *S. aureus* FtsZ and obtained the best scores in the interdomain site, with strikingly similar poses to the binding mode of PC190723 [66]. The two molecules have similar planarity, shape, length, and alignment of the ring system. Berberine established extensive hydrophobic interactions with nearby residues Ile197, Leu200, Val203, Leu209, Met226, Leu261, Val297, and Ile311, while the positively charged amine was found to interact with Asp199. The model indicated the possibility of additional hydrophobic interactions through lipophilic substituents in position C-9. Propyl 4-chlorophenoxy compound **I16** (Figure 8) was the most potent among the tested derivatives, with greatly increased antimicrobial potency against a panel of clinically relevant drug-resistant Gram-positive and Gram-negative strains compared to berberine (Table 8). The inhibitory mechanism on FtsZ is supported by the inhibition of *S. aureus* FtsZ GTPase activity (IC_50_ = 37.8 μM) and by the inhibition of *S. aureus* FtsZ polymerization (~70%) in light scattering assays. TEM visualization of *S. aureus* FtsZ protofilaments in vitro showed reduced size, thickness, and bundling. **I16** induces filamentation of *B. subtilis* and causes dispersion and mis-localization of GFP-tagged *E. coli* FtsZ. Additional studies on berberine analogues and, more specifically, on the SAR of C-13 substituted cycloberberines led to the development and characterization of **I17** (Figure 8). Despite its promising antistaphylococcal activity, with MICs of 1.8–7.4 μM against 10 MSSA and MRSA strains (including drug-resistant clinical isolates), **I17** has critical pharmacokinetic issues, specifically regarding oral bioavailability (0.8% in mice). In simulations, **I17** docked efficiently at the interdomain site, with the extra *o*-methylbenzyl ring engaged in additional hydrophobic interactions. 

Prompted by the interesting antimicrobial activity of berberine and its analogues, Sun and coworkers explored the potential and the structure–activity relationships of the quinolinium scaffold in a recent series of studies.

Interestingly, this group of structurally related compounds appear to act through two opposite mechanisms, being either inhibitors or enhancers of FtsZ polymerization activity. **I18** [67] is the most potent in a series of quaternary *N*-methylbenzoindolo[3,2-b]-quinoline analogues, with good bactericidal activity on both MSSA and MRSA (MIC = 4.1 μM), vancomycin-sensitive and -resistant *E. faecium* (MIC = 8.3 μM), and *E. coli* (MIC = 12.5 μM, including a beta-lactamase producing strain). In addition, it shows moderate activity on drug-resistant strains of Gram-negative *P. aeruginosa* and *K. pneumoniae* (MIC = 100 μM), in clear contrast to the very weak activity of berberine (MIC > 400 μM). Docking simulations suggested binding in the interdomain cleft, favored by hydrophobic interactions and by the ionic interaction of the positive nitrogen with negatively charged Asp199. The planarity of the molecule may represent a key factor for accessibility to the narrow space of the binding site and, given the inter-species differences in the opening size, may be determinant for broad-spectrum activity. Moreover, the indole nitrogen can form additional hydrogen bonding with the backbone carbonyl of Thr309, providing an explanation for the better activity of **I18** compared to its benzofuroquinolinium counterpart. The inhibition of both GTPase activity and polymerization was demonstrated with the well-established combination of GTPase assays and light scattering polymerization assays on *S. aureus* FtsZ, confirming the on-target mechanism of action. The typical filamentation of treated *B. subtilis* further supports the inhibitory action of FtsZ. Based on a benzofuroquinolinium scaffold, **I19** [68] shares the same mechanism of action as its predecessor, with similar activity on *Staphylococcii* (MIC = 1.5–3 μM), *Enterococcii* (MIC = 3–6 μM), and *E. coli* (MIC = 6 μM) and improved potency on Gram-negative *P. aeruginosa*, *K. pneumonia*, and *A. baumannii* (MIC = 25 μM). In addition, it appears to restore the sensitivity of MRSA to methicillin and to be devoid of toxicity on mammalian cells and human erythrocytes. **I20 [50]** and its *p*-hydroxy analogue **I21 [51]** further reduce the activity gap between Gram-positive and Gram-negative strains, with MICs as low as 6 μM on *E. coli* (including a MDR strain) and 12 μM on MDR *P. aeruginosa* (Table 8). The predicted binding mode on *S. aureus* FtsZ involves docking in the interdomain cleft, with a network of hydrophobic and ionic interactions. An additional hydrogen bond can be found between Asp199 and the hydrogen of the methyl group, together with van der Waals interactions with a few nearby residues (e.g., Gln192 and Gly196). The hydroxyl of **I21** is predicted to form two hydrogen bonds: with the backbone carbonyl of Val203 and with the backbone amide of Leu209. The extended SAR for the 3-methylbenzo[d]thiazol-methylquinolinium is reported in a distinct paper, with no substantial improvements in antibacterial activity [69]. The same authors synthesized and tested a library of indolylquinolinium analogues [70], with the most potent compound (**I22**) showing no significant improvement in the activity on Gram-positive strains and marked reduction of potency on Gram-negative strains, in some cases with MIC > 130 μM (Table 8). **I20**, **I21**, and **I22** share the same bactericidal mechanism of action, as shown by GTPase and polymerization activity assays. All three compounds disrupt GTP hydrolysis in a concentration-dependent manner while enhancing the formation of FtsZ protofilaments. Therefore, they act as polymer-stabilizing agents, most likely through the promotion of a highly elongation-prone FtsZ conformer, in a similar way to that observed for PC190723 and related benzamides.

#### 2.2.3. Phenantridium Derivatives

Sanguinarine (**I23** in Figure 9) is a benzofenantridium plant alkaloid with antimicrobial activity, able to perturb the formation of the Z−ring and to cause filamentation of *B. subtilis* and *E. coli*. It inhibits the assembly of FtsZ and reduces the bundling of protofilaments in vitro [71]. However, it also inhibits the polymerization of tubulin into microtubules and it is toxic to mammalian cells [72]. Liu et al. aimed at developing non−toxic and potent antimicrobials from the structural simplification of sanguinarine. The 5−methylphenantridium derivative **I24 [73]** shows moderate activity against Gram−positive bacteria and substantial inactivity against Gram−negative ones (Table 9), with an overall activity profile comparable to parent molecule **I23**. On the other hand, when tested for their effect on mammalian tubulin, none of the analogues enhanced polymerization at concentrations higher than positive control paclitaxel. The evaluation of on−target effects with specific assays and the determination of toxicity on mammalian cells are not reported, and the potential of **I23** and congeners as novel FtsZ inhibitors is difficult to evaluate with current data. As a continuation of their work on the 5−methylphenantridium scaffold, the same authors synthesized and tested three series of 4−substituted 5−methyl−2−phenylphenantridium analogues. The effect of substitution on C−4 was generally positive, with 4−methyl derivatives being more active than 4−methoxy and 4−unsubstituted ones. Additionally, the effect of substitutions on the phenyl ring was evaluated, with a clear indication that electron−withdrawing groups are favored over electron−donating ones. Accordingly, **I25 [52]** is the most potent compound in the series, with outstanding MICs of 0.16–5.2 μM on various drug−sensitive and −resistant strains of Gram−positive bacteria (Table 9), performing much better than reference compounds ciprofloxacin and oxacillin. Kinetic inhibition profiles indicated a bactericidal effect. Microscopic morphometric evaluation showed elongation and swelling of treated *B. subtilis* and *S. aureus*, respectively, consistent with the disruption of cell division processes. Light scattering assays displayed concentration−dependent reduction of *B. subtilis* FtsZ polymerization activity. Taken together, this evidence supports the on−target mechanism of action of **I25** as FtsZ inhibitor. Molecular docking simulations suggested the binding of **I25** in the interdomain cleft of FtsZ, favored by extensive hydrogen bond and hydrophobic interactions.

#### 2.2.4. Indoles (Tiplaxtinin)

In a cell−based screening for morphology alterations in *B. subtilis*, Sun et al. identified tiplaxtinin (**I26** in Figure 10) as a promising hit, out of a library of 150 small molecule candidates. Tiplaxtinin is already known for being an efficacious plasminogen activator inhibitor−1 (PAI−1) inhibitor and a potential antithrombotic agent, with no reported toxicity and good oral availability in mice. It displays good antimicrobial potency on a group of Gram−positive bacteria, with MICs of 4.55–9.10 µM, despite substantial inactivity on *E. coli*, *P. aeruginosa*, and *K. pneumoniae* (MIC > 109 µM) (Table 10). The on−target activity of **I26** on *S. aureus* FtsZ was thoroughly assessed through light scattering polymerization assays, GTPase assays, and TEM microscopy, revealing concentration−dependent enhancement of polymerization, a decrease in GTPase activity, and a significant increase in protofilament bundling in the presence of the compound. Moreover, **I26** induces the mis−localization of GFP−tagged FtsZ in discrete and punctate foci, indicating the mis−formation of the Z−ring [53]. Docking simulations predicted the binding of **I26** in the interdomain cleft, supported by hydrophobic interactions in the lipophilic channel and by a set of additional interactions with specific functional groups. The good FtsZ inhibitory activity, together with the available knowledge on toxicity and oral bioavailability, makes the tiplaxtinin scaffold a promising candidate for further structural optimization and for the characterization of its efficacy in in vivo models of infection.

## 3. Conclusions

Some interesting trends emerged from the presented data; here, we summarize them, hoping that these conclusions could be useful for the further development of FtsZ inhibitors as antimicrobial agents.

Concerning interspecies susceptibility, strong differences were reported even among compounds having the same or similar chemotypes. Most of the presented classes appear to predominantly affect Gram−positive strains, while having scarce to null effects on the inhibition of FtsZ in Gram−negative species.

We noticed a strict classification of some of these compounds as “inactive on Gram−negative strains” that may be misleading, as the overall view of this topic is much more complicated and diversified. Indeed, a progressive reduction of the activity gap between Gram−positive and Gram−negative strains during the sequential optimization of quinolinium compounds was reported. On the contrary, the structural simplification of the quinuclidine scaffold of **N2** to more flexible amines caused the loss of Gram−negative inhibitory activity. A similar outcome was also observed in *S. aureus*, considering the activity of flexible analogue **I8** on resistant *S. aureus* strains. In our opinion, these results suggest that an efficient structural optimization could balance the differences in susceptibility between Gram−positive and Gram−negative bacteria.

Several factors seem to play a role in the determination of susceptibility: (a) the spatial characteristics of the FtsZ binding sites across different species, independently of the Gram staining, as exemplified by the inactivity of some benzamide inhibitors (notably, PC190723) on Gram−positive *Enterococcii* and *Streptococcii*; (b) the differences in the amino−acidic sequence of the binding sites (especially for the interdomain binding site) across different species; (c) being a substrate for the most common membrane efflux pumps, as exemplified by the partial recovery of activity of two benzamide−based inhibitors (**I4** and **I10**) and pyrimidine **N4** after the genetic or pharmacological deactivation of AcrAB. 

Moreover, we noticed an overall imbalance in the number and in the variety of Gram−positive strains vs. Gram−negative strains tested during antimicrobial activity assays, with a clear abundance of the former, specifically *Staphylococcii*. Even if this disproportion could be related to the most interesting initial antimicrobial results and to the availability of crystal structures for rational design, we believe that a more balanced approach could be beneficial for a more exhaustive study of the topic.

Furthermore, the variety of Gram−negative strains tested was usually limited to *E. coli*, *K. pneumoniae* (both expressing AcrAB), and *P. aeruginosa*, which have an extensive array of cromosomically encoded RND efflux pumps, very limited membrane permeability, and a known intrinsic non−susceptibility to several antimicrobials. We believe that the previously mentioned results with efflux pump inhibitors (EPIs) stress the potential of using FtsZ inhibitors in broad−spectrum associations, as well as of evaluating the susceptibility of both existing and newly developed leads to efflux pumps. Indeed, a growing knowledge on FtsZ inhibitors could be a highly valuable resource for their future development, given the clinical relevance and the generally limited therapeutic options available for Gram−negative bacteria.

## Figures and Tables

**Figure 1 antibiotics-09-00069-f001:**
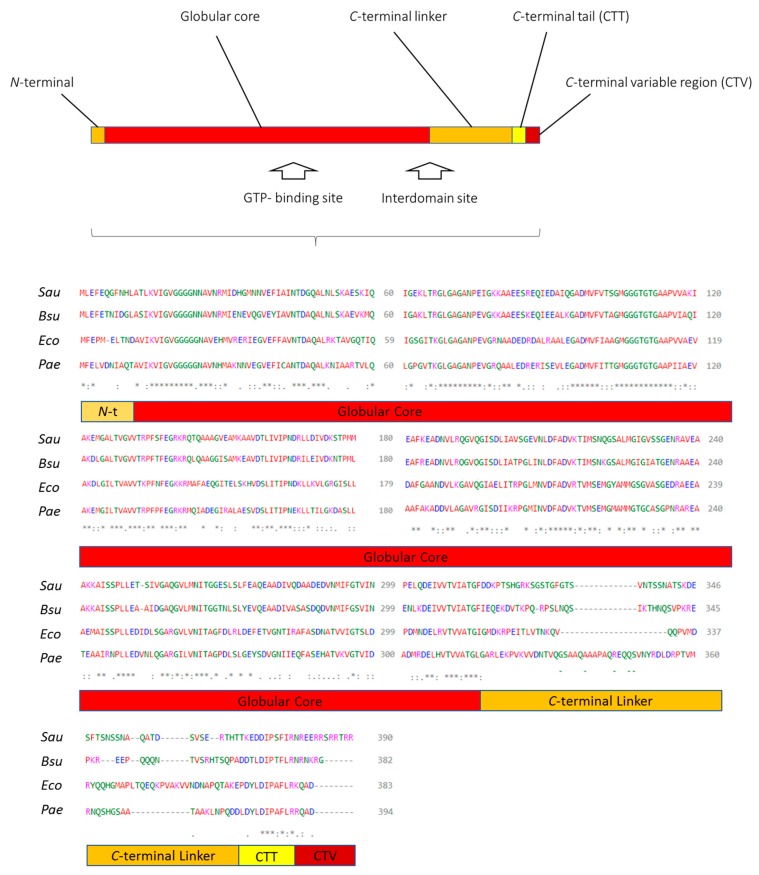
Graphical representation of FtsZ main subunits and sequence alignment of FtsZ from representative organisms: *Staphylococcus aureus* (*Sau*), *Bacillus subtilis* (*Bsu*), *Escherichia coli* (*Eco*), and *Pseudomonas aeruginosa* (*Pae*). Alignments: * = same residues, : = equivalent residues, · = partial alignment. The sequences were obtained from uniport.org (identifiers (IDs): P0A031, P17865, P0A9A6, P47204) and aligned with EMBL-EBI Clustal Omega (www.ebi.ac.uk/Tools/msa/clustalo/).

**Figure 2 antibiotics-09-00069-f002:**
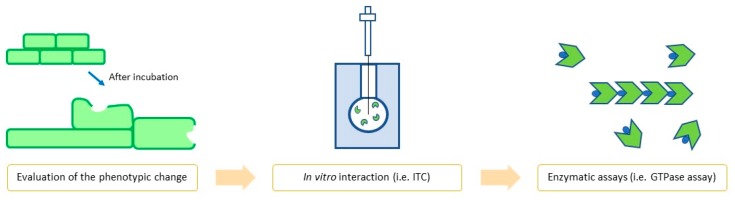
Flowchart of the main steps in FtsZ inhibition confirmation.

**Figure 3 antibiotics-09-00069-f003:**
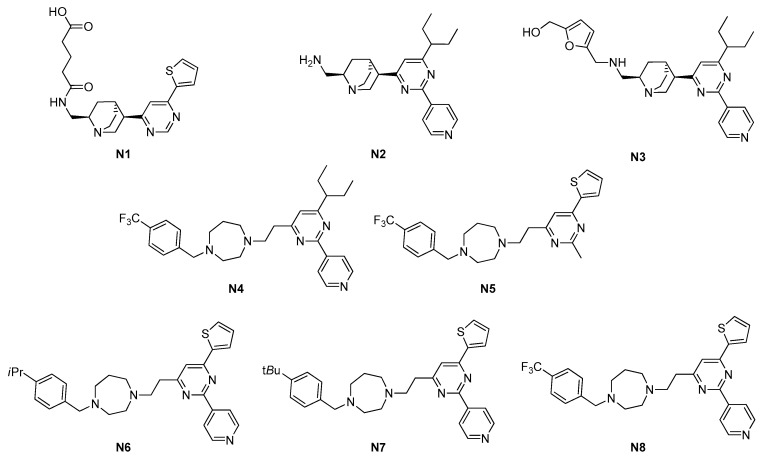
Pyrimidine FtsZ inhibitors.

**Figure 4 antibiotics-09-00069-f004:**
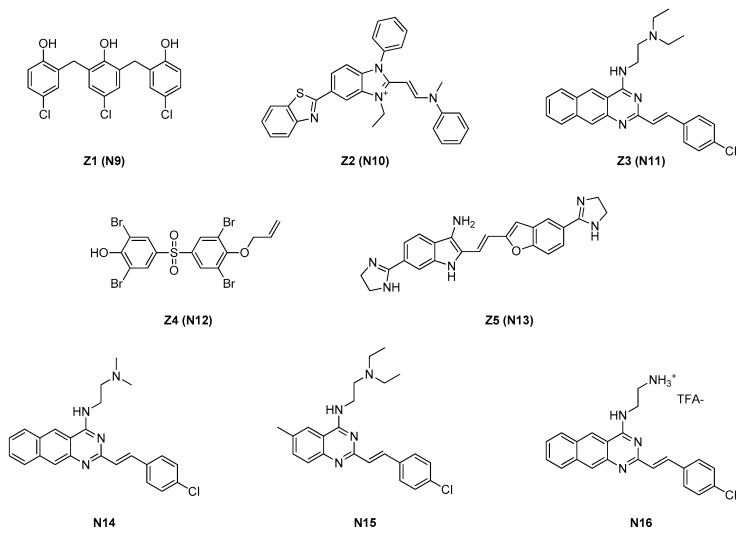
Zantrin FtsZ inhibitors.

**Figure 5 antibiotics-09-00069-f005:**
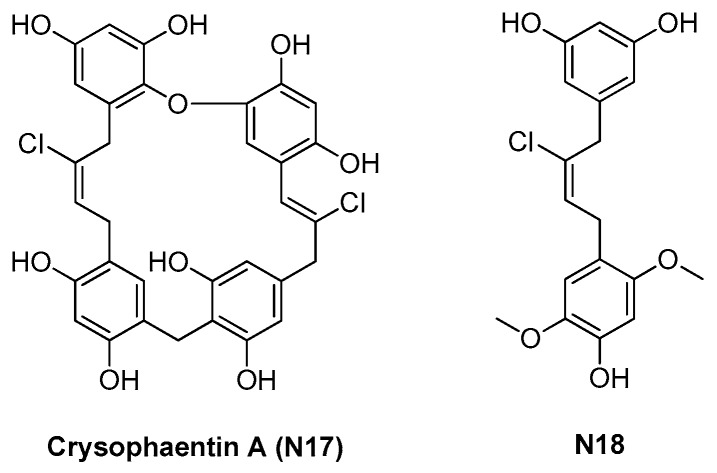
Chrysophaentin FtsZ inhibitors.

**Figure 6 antibiotics-09-00069-f006:**
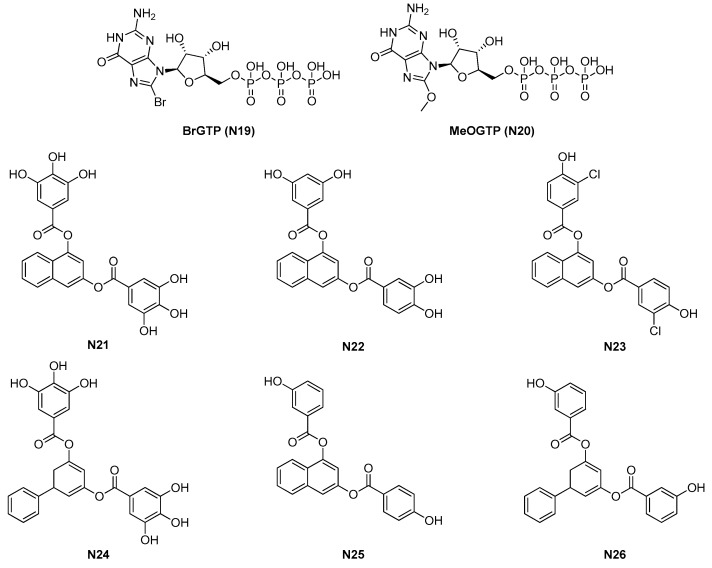
GTP analogues and derived synthetic FtsZ inhibitors.

**Figure 7 antibiotics-09-00069-f007:**
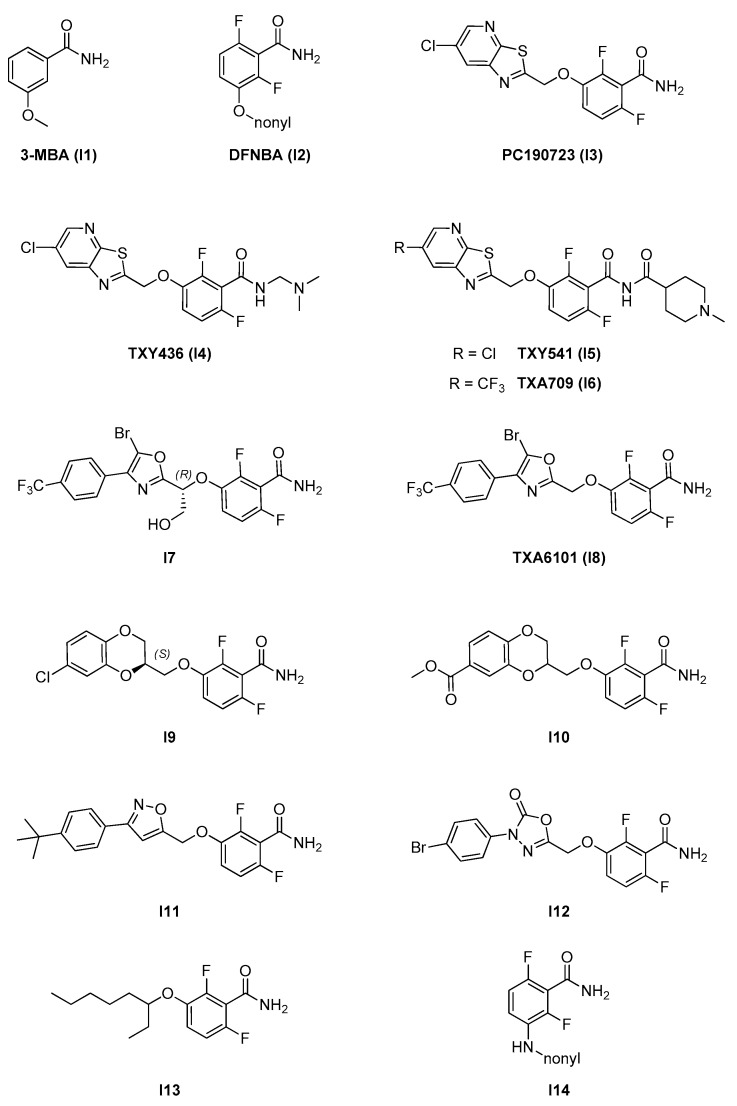
Benzamide inhibitors.

**Figure 8 antibiotics-09-00069-f008:**
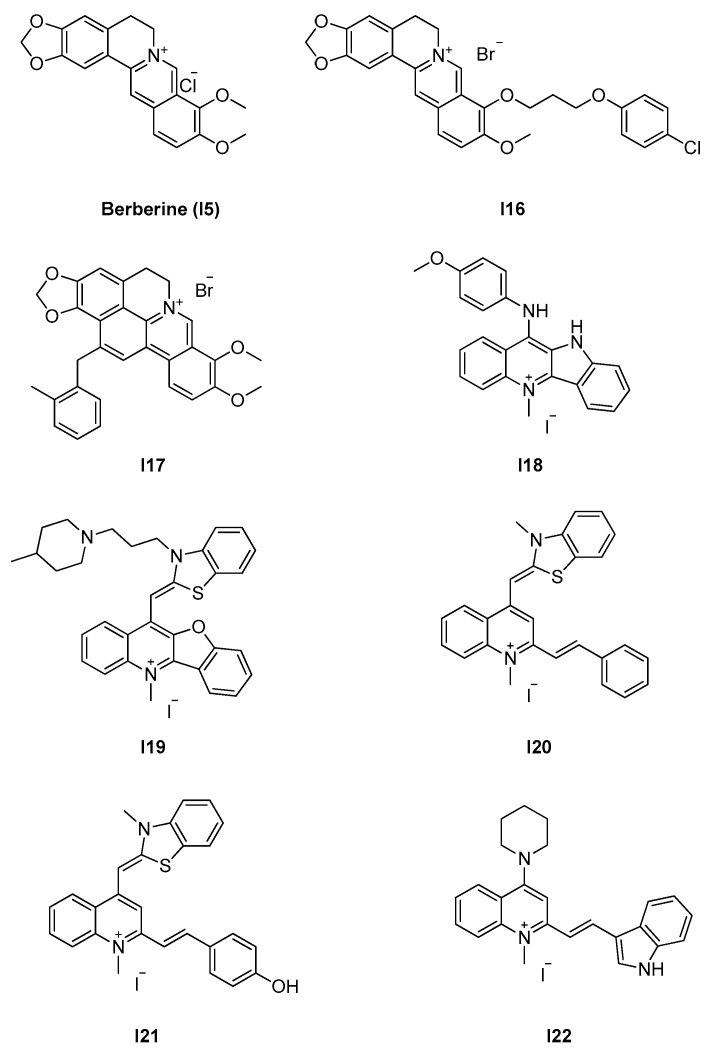
Berberine-based inhibitors.

**Figure 9 antibiotics-09-00069-f009:**
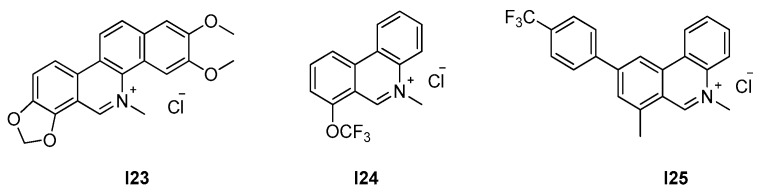
Phenantridium inhibitors.

**Figure 10 antibiotics-09-00069-f010:**
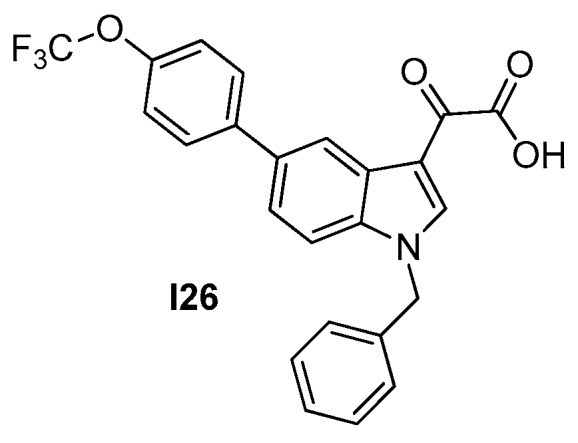
Tiplaxtinin.

**Table 1 antibiotics-09-00069-t001:** General overview of the best FtsZ inhibitors. MIC—minimum inhibitory concentration; IC_50_—half maximal inhibitory concentration; STD—saturation transfer difference; ITC—isothermal titration calorimetry.

Site	Class *(Figure)*	Best Compounds	MICs	Evidence for FtsZ Inhibition	References
2.1. GTP-binding site	2.1.1. Pyrimidines (*3*)	**N6**–**N8**	*S. aureus*: 4–8 μM *Enterococcus faecalis*: 4–8 μM *E. faecium*: 4–8 μM	Polymerization assay; GTPase activity assay; Microscopy evaluation.	[14,15]
	2.1.2. Zantrins (*4*)	**N11**	*S. aureus*: 5 μM *E. coli:* DRC39: 10 μM	IC_50_ evaluation; Molecular modeling.	[16,17,18,19]
	2.1.3. Chrysophaentins (*5*)	**N17**	*S. aureus*: 5–9 μM *E. faecium*: 5–6 μM	Polymerization assay; GTPase activity assay; STD-NMR; Molecular modeling.	[20,21,22]
	2.1.4. GTP analogues and derivatives (*6*)	**N23-N26**	*B. subtilis*: 8.5 μM *S. aureus:* 5–50 μM	Microscopy evaluation; Sedimentation assay; Polymerization assay; Molecular modeling.	[23,24]
2.2. Interdomain site	2.2.1. Benzamides (*7*)	**I3; I4; I7**–**I12**	*B. subtilis*: <1 μM *S. aureus:* 1.6–2.8 μM *E. coli* N43: 19–42 μM	Polymerization assay; GTPase activity assay; Microscopy evaluation; Molecular modeling; X-ray crystallography.	[25,26,27,28,29,30,31,32,33,34,35,36,37,38,39,40,41,42,43,44,45,46,47,48,49]
	2.2.2. Berberine analogues and derivatives (*8*)	**I20**–**I22**	*B. subtilis*: 2.8–8 μM *S. aureus:* 2.7–8 μM *E. coli*: 5.5 μM	Polymerization assay; GTPase activity assay; Microscopy evaluation; STD-NMR; ITC.	[50,51]
	2.2.3. Phenantridium derivatives (*9*)	**I25**	*B. subtilis*: 12.7 μM *S. aureus:* 51 μM	Microscopy evaluation; Polymerization assay; Molecular modeling.	[52]
	2.2.4. Indoles (*10*)	**I26**	*B. subtilis*: 4.5 μM *S. aureus:* 4.5–9.1 μM	Polymerization assay; GTPase activity assay; Microscopy evaluation.	[53]

**Table 2 antibiotics-09-00069-t002:** MICs of Pyrimidines N1–N8. ATCC—American Type Culture Collection.

Strain	MIC (μM)							
	N1	N2	N3	N4	N5	N6	N7	N8
*S. aureus* ATCC 29213	897.1	24.6	50	8	52	4	4	8
*S. aureus* ATCC 29247	−	−	50	5.8	52	−	−	−
*S. aureus* ATCC BAA-1717 *	−	−	−	5.8	52	−	−	−
*S. aureus* ATCC BAA-1720 *	−	−	−	11.7	52	4	4	8
*S. aureus* ATCC BAA-41 *	−	−	50	11.7	52	4	4	8
*S. aureus* ATCC BAA-1747 *	−	−	−	−	52	−	−	−
*S. aureus* ATCC 43300 *	−	−	−	11.7	−	4	4	8
*S. aureus* USA300 #757	−	−	−	11.7	−	−	−	−
*S. aureus* USA300 #1799	−	−	−	11.7	−	−	−	−
*S. aureus* USA300 #2690	−	−	−	11.7	−	−	−	−
*S. aureus* ATCC 33591 *	−	−	−	−	−	4	4	8
*S. aureus* ATCC 33592 *	−	−	−	−	−	4	4	8
*B. subtilis* 168	−	−	50	−	52	4	4	8
*E. faecalis* ATCC 29212	−	−	50	−	208	4	4	8
*E. faecalis* ATCC 51575 **	−	−	−	−	208	4	8	8
*E. faecium* ATCC 49624	−	−	−	−	208	4	8	8
*E. faecium* ATCC 700221 **	−	−	−	−	208	4	8	8
*S. epidermidis* ATCC 12228	−	−	−	−	−	4	4	8
*E. coli* ATCC 25922	449.0	49.2	76	>64	208	>208	>208	>208
*E. coli* ATCC BAA-2469	−	−	−	−	208	−	−	−
*P. aeruginosa* ATCC BAA-2108	−	−	−	−	>208	−	−	−
*Klebsiella pneumoniae* ATCC BAA-1144	−	−	−	−	>208	−	−	−

* methicillin-resistant *S. aureus* (MRSA), ** vancomycin-resistant.

**Table 3 antibiotics-09-00069-t003:** MICs of Zantrins N9–N13.

Strain	MIC (μM)				
	N9	N10	N11	N12	N13
*S. aureus* H	2.5	1.25	5	10	>80
*S. aureus* clinical MRSA	2.5	2.5	10	10	>80
*Streptococcus pneumoniae* TIGR 4	0.3	2.5	5	10	>80
*Clostridium perfringens* Strain 13	5	10	80	5	>80
*B. subtilis* 168	1.25	2.5	2.5	2.5	2.5
*B. cereus* CIP 3852	0.6	5	20	2.5	2.5
*E. coli* MC 1000	20	40	>80	>80	>80
*E. coli* DRC 39	20	5	10	>80	80
*Shigella dysenteriae* 60R	10	10	20	>80	>80
*Vibrio cholerae* N16961	5	5	5	>80	>80
*P. aeruginosa* PAK	40	>80	>80	>80	>80

**Table 4 antibiotics-09-00069-t004:** MICs of chrysophaentins N17–N18.

Strain	MIC (μM)	
	N17	N18
*S. aureus* 25293	9.2	34
*S. aureus* MRSA BAA-41	4.6	34
*S. aureus* MDRSA BAA-44	9.2	34
*S. aureus* UAMS-1	9.2	34
CA-MRSA USA 300-LAC	9.2	34
*E. faecium **	6.1 *	−
*E. faecium* VREF *	4.6 *	−

* Expressed as MIC_50._

**Table 5 antibiotics-09-00069-t005:** MICs of non-nucleotide GTP analogues N21–N26.

Strain	MIC (μM)					
	N21	N22	N23	N24	N25	N26
*B. subtilis* 168	>100	38	8.5		−	−
*S. aureus* ATCC 29213	69	38	17	−	−	−
*S. aureus* 12160636 *	69	38	17	−	−	−
*S. aureus* MDRSA Mu50	80	−	−	50	5	7
*E. faecium* 12160560 **	>100	74	8.5	−	−	−
*E. faecalis* ATCC 29212	>100	74	4.3	−	−	−
*E. faecalis* 12165475 *****	>100	74	8.5	−	−	−
*E. faecalis* V583 ****	>100	−	−	>100	50	50
*S. pneumoniae* ATCC 49619	138	74	68.2	>100	74	68.2
*P. aeruginosa* ATCC 27853	>100	>100	>100	50	>100	>100
*K. pneumoniae* ATCC 700603	>100	>100	>100	>100	>100	>100
*E. coli* ATCC 35218	>100	>100	>100	50	>100	>100

* MRSA, ** multidrug-resistant (MDR), *** levofloxacin-resistant.

**Table 6 antibiotics-09-00069-t006:** MICs of benzamides I3–I6.

Strain	MIC (μM)			
	I3	I4	I5	I6
*S. aureus* ATCC 29213	2.81	2.42	-	3.89
*S. aureus* ATCC 19636	2.81	1.21	4.15	3.89
*S. aureus* ATCC 43300 *	2.81	1.21	4.15	7.77
*S. aureus* ATCC BAA-44 **	2.81	−	−	−
*S. aureus* 8325-4	1.40	1.21	4.15	3.89
*S. aureus* ATCC 49951	1.40	2.42	−	−
*S. aureus* ATCC 33591	1.40	1.21	4.15	7.77
*S. epidermidis* ATCC 12228	2.81	−	−	−
*S. haemolyticus* ATCC 29970	1.40	−	−	−
*S. hominis* ATCC 27844	2.81	−	−	−
*S. lugdunensis* ATCC 43809	2.81	−	−	−
*S. saprophyticus* ATCC 15305	2.81	−	−	−
*S. warneri* ATCC 49454	2.81	−	−	−
*B. cereus* ATCC 14579	2.81	−	−	−
*B. subtilis* 168	2.81	−	−	−
*S. pneumoniae* ATCC 49619	>180	−	−	−
*S. pyogenes* ATCC 51339	>180	−	−	−
*S. pyogenes* ATCC 19615	>180	−	−	−
*S. agalactiae* ATCC 12386	>180	−	−	−
*E. faecalis* ATCC 19433	90.0	−	−	−
*E. faecalis* ATCC 51575 ***	90.0	−	−	−
*E. faecium* ATCC 19434	180	−	−	−
*E. coli* ATCC 25922	>180	>155	−	−
*E. coli* N43	−	19.4	−	−
*Haemophilus influenzae* ATCC 49247	>180	−	−	−
*P. aeruginosa* ATCC 27853	>180	−	−	−
*K. pneumoniae* ATCC 13883	−	>155	−	−
*Acinetobacter baumannii* ATCC 19606	−	>155	−	−

* MRSA, ** multidrug-resistant *S. aureus* (MDRSA). *** multidrug-resistant *E. faecalis*.

**Table 7 antibiotics-09-00069-t007:** MICs of benzamides I7–I13.

Strain	MIC (μM)						
	I7	I8	I9	I10	I11	I12	I13
*S. aureus* ATCC 29213	2.81	2.42	0.53	1.58	−	−	28.0
*S. aureus* ATCC 19636	2.81	1.21	−	−	−	−	−
*S. aureus* ATCC 43300 *	2.81	1.21	1.10	−	5.17	2.34	−
*S. aureus* ATCC 25923	−	−	−	−	5.17	2.34	3.50
*S. aureus* ATCC BAA−44 **	2.81	−	−	−	−	−	−
*S. aureus* 8325-4	−	1.21	−	−	−	−	−
*S. aureus* ATCC 49951	−	2.42	−	−	−	−	−
*S. aureus* ATCC 33591	−	1.21	−	−	−	−	−
*S. epidermidis* ATCC 12228	2.81	−	−	−	−	−	−
*S. haemolyticus* ATCC 29970	1.40	−	−	−	−	−	−
*S. hominis* ATCC 27844	2.81	−	−	−	−	−	−
*S. lugdunensis* ATCC 43809	2.81	−	−	−	−	−	−
*S. saprophyticus* ATCC 15305	2.81	−	−	−	−	−	−
*S. warneri* ATCC 49454	2.81	−	−	−	−	−	−
*B. cereus* ATCC 14579	2.81	−	−	−	−	−	−
*B. subtilis* 168	2.81	−	−	−	−	−	−
*B. subtilis* ATCC9372	−	−	−	−	0.04	0.29	0.87
*B. pumilus* CMCC63202	−	−	−	−	0.08	2.34	−
*S. pneumoniae* ATCC 49619	>180	−	−	−	−	−	224
*S. pyogenes* ATCC 51339	>180	−	−	−	−	−	−
*E. coli* ATCC 25922	>180	>155	−	−	>165	>150	448
*E. coli DH5α*	−	−	>281	−	−	−	−
*E. coli* D22	−	−	−	>337	−	−	−
*E. coli* N43	−	−	−	42.2	−	−	−
*H. influenzae* ATCC 49247	>180	−	−	−	−	−	−
*P. aeruginosa* ATCC 27853	>180	−	−	−	>165	>150	897
*K. pneumoniae* ATCC 13883	−	>155	−	−	−	−	−
*A. baumannii* ATCC 19606	−	>155	−	−	−	−	−

* MRSA, ** MDRSA.

**Table 8 antibiotics-09-00069-t008:** MICs of berberines I15–I22.

Strain	MIC (μM)							
	I15	I16	I17	I18	I19	I20	I21	I22
*S. aureus* ATCC 29213	360	3.50	7.35	4.15	3.09	2.80	2.72	2.02
*S. aureus* ATCC 25923	−	−	−	−	3.09	−	−	4.04
*S. aureus* ATCC 29247 *	360	7.00	−	−	−	2.80	−	−
*S. aureus* ATCC BAA-41 **	551	7.00	−	4.15	3.09	2.80	5.45	4.04
*S. aureus* ATCC 33591 **	−	−	3.67	−	3.09	−	5.45	−
*S. aureus* ATCC 33592 **	−	−	−	−	3.09	−	−	−
*S. aureus* ATCC 43300 **	−	−	3.67	−	3.09	−	2.72	8.07
*S. aureus* ATCC BAA-1717 **	−	−	−	−	3.09	2.80	−	−
*S. aureus* ATCC BAA-1720 **	−	−	−	−	3.09	2.80	−	−
*S. aureus* ATCC BAA-1747 **	−	−	−	−	−	2.80	−	−
*S. aureus* ATCC BAA-976 **	−	−	3.67	−	−	−	−	−
*S. aureus* ATCC BAA-1708 **	−	−	7.35	−	−	−	−	−
*S. epidermidis* ATCC 12228	360	3.50	−	−	1.54	1.40	1.36	−
*E. faecium* ATCC 49624	>551	7.00	−	8.31	3.09	3.74	1.36	4.04
*E. faecium* ATCC 700221 ***	>551	7.00	−	8.31	3.09	3.74	2.72	8.07
*E. faecalis* ATCC 29212	>551	7.00	−	−	6.17	2.80	1.36	8.07
*E. faecalis* ATCC 51575	−	−	−	−	6.17	2.80	−	−
*B. subtilis* 168	360	7.00	−	4.15	0.77	2.80	2.72	8.07
*E. coli* ATCC 25922	>1405	56.0	−	12.4	6.17	5.61	5.45	>129
*E. coli* ATCC BAA−2469 ^§^	−	−	−	12.4	6.17	5.61	5.45	>129
*P. aeruginosa* ATCC BAA-2108 ^§§^	−	−	−	100	24.7	11.2	10.9	>129
*K. pneumoniae* ATCC BAA-2470 ^§^	−	−	−	100	−	−	−	>129
*K. pneumoniae* ATCC BAA-1144 ^§^	>1405	112	−	−	24.7	44.9	87.2	−
*A. baumannii* ATCC 19606 ^§§^	−	−	−	−	24.7	−	87.2	>129
*Enterobacter cloacae* BAA-1143 ^§^	−	−	−	−	−	−	−	>129

*** Ampicillin−resistant, ** methicillin resistant, *** vancomycin−resistant, ^§^ expresses beta−lactamases, ^§§^ MDR.

**Table 9 antibiotics-09-00069-t009:** MICs of phenantridium inhibitors I23–I25.

Strain	MIC (μM)		
	I23	I24	I25
*S. aureus* ATCC 29213	20.8	−	−
*S. aureus* ATCC 25923	20.8	51	0.15
*S. aureus* ATCC 43300 **	−	−	0.64
*S. epidermidis **	166	102	0.32
*S. pyogenes*	20.8	12.7	2.58
*S. pyogenes **	20.8	25.5	5.16
*B. subtilis* ATCC 9372	20.8	12.7	0.15
*B. pumilus* ATCC 63202	−	−	0.15
*E. coli* ATCC 25922	>333	>408	−
*P. aeruginosa* ATCC 27853	>333	>408	−

* Penicillin−resistant, ** methicillin−resistant.

**Table 10 antibiotics-09-00069-t010:** MICs of Tiplaxtinin (I26).

Strain	MIC (μM)
	I26
*S. aureus* ATCC 29213	4.55
*S. aureus* ATCC 29247 *	4.55
*S. aureus* ATCC BAA−41 **	9.10
*S. aureus* ATCC 33591 **	9.10
*S. aureus* ATCC 33592 **	9.10
*S. aureus* ATCC 43300 **	9.10
*S. aureus* ATCC BAA−41 **	9.10
*S. aureus* ATCC BAA−1717 **	4.55
*S. aureus* ATCC BAA−1720 **	4.55
*S. aureus* ATCC BAA−1747 **	4.55
*E. faecium* ATCC 49624	9.10
*E. faecium* ATCC 700221 ***	9.10
*E. faecalis* ATCC 29212	910
*B. subtilis* 168	4.55
*E. coli* ATCC 25922	>109
*P. aeruginosa* ATCC BAA−2108 ^§§^	>109
*K. pneumoniae* ATCC BAA−1144 ^§^	>109

*** Ampicillin−resistant, ** methicillin−resistant, *** vancomycin−resistant, ^§^ expresses beta−lactamases, ^§§^ MDR.

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
