# Peer review of "Targeting Bacterial Cell Division: A Binding Site-Centered Approach to the Most Promising Inhibitors of the Essential Protein FtsZ"

_antibiotics, 2020, doi:10.3390/antibiotics9020069_

Round 1

Reviewer 1 Report

This paper reviews recent advances in developing FtsZ inhibitors, with reference to their binding sites, and with a medicinal chemistry focus. I found it interesting and useful for catching up on recent developments in the field.

My feeling is that this manuscript needs work with respect to editing and usage of English. For instance, the first sentence of the abstract is awkward and I would re-write it as follows:

“Binary fission is the most common mode of bacterial cell division, and it is mediated by a multiprotein complex denominated divisome. The constriction of the Z-ring splits the mother bacterial cell into two daughter cells of the same size.”

Therefore I would accept for publication after editing for English usage.

Author Response

Reviewer 1: We would thank the reviewer for all the positive comments on the overall contribution to the field, on the paper organization and on the number of references.

We corrected all the grammatical errors, the mistakes and we revised the manuscript, especially in the usage of English. We take advantage from the help of an English native speaker.

Reviewer 2 Report

The authors have written a manuscript on inhibitors of Ftsz, a protein essential for bacterial cell division. The manuscript has lot of details and is quite comprehensive. It will be useful if the authors can better organize all the text and provide extra figures to explain the text. I have few suggestions that are listed below: 

The introduction needs to be revised to include some broader details on cell division. The manuscript needs to start with a broader explanation of cell division before going into details about Ftsz  Figure 1 can show the Ftsz protein in other organisms too to explain the conserved regions. Also it will be useful to include what proteins bind to which regions in the protein. This should also be explained in the text  The many functions of Ftsz or how it works should be briefly described in the introduction. This is especially important because later in the text, the authors discuss how the inhibitors affect Ftsz activity  A cartoon or table that lists the various studies discussed and the compounds discovered (including significant details of specific compounds) will be useful to readers 

Author Response

Reviewer 2: The reviewer underlined a few criticisms, concerning the general organization of the text (point 1), the explanation of the cell division process and the FtsZ functions (point 2) and the summary of the best FtsZ inhibitors (point 3).

We appreciate all the sharp and accurate comments from the reviewer, inciting us to implement the manuscript, and we introduced the following modifications:

We deeply revised the introduction, correcting all the grammar mistakes, editing the English usage, deleting a useless subchapter and adding the general chapter of the FtsZ inhibitors (chapter 2). The help of an English native speaker let us to have a fluent overall work. The introduction was implemented in all its subchapters, especially in 1.1, where we better elucidated the cell division process, we introduced the main proteins involved and we stressed the importance of all the FtsZ functions.

Figure 1 was changed, introducing the comparison between the sequences of 4 different FtsZ, belonging to the most tested strains, 2 Gram-positive and 2 Gram-negative. We think that such a comparison could be very useful for the readers in order to understand the different degree of conservation of each single domain. Furthermore, we also highlighted, in the graphical representation of FtsZ, the two main binding sites of the FtsZ inhibitors.

The second and third chapters were merged, generating a unique general chapter, where we introduced the main FtsZ inhibitor classes and we included a summary table (Table 1). As the reviewer suggested, we think this summary table could be very useful to readers in understanding the most significant studies, together with the best FtsZ inhibitor candidates and their details.

Moreover, we checked the whole review, better equalizing all the tables and the overall text formatting.